# SARS-CoV-2 Emergency Management in the ASL 1 Abruzzo Companies, Italy: An Autumn 2022 Cross-Sectional Investigation

**DOI:** 10.3390/ijerph20065105

**Published:** 2023-03-14

**Authors:** Marianna Mastrodomenico, Maria Grazia Lourdes Monaco, Antonio Spacone, Enrica Inglese, Arcangelo Cioffi, Leila Fabiani, Elpidio Maria Garzillo

**Affiliations:** 1Corporate Service of Penitentiary Medicine, Abruzzo Local Health Unit No.1, 67100 L’Aquila, Italy; 2Occupational Medicine Unit, University Hospital of Verona, 37134 Verona, Italy; 3Department of Prevention, Abruzzo Local Health Unit No.1, 67100 L’Aquila, Italy; 4Section of Occupational Medicine, Department of Diagnostics and Public Health, University of Verona, 37134 Verona, Italy; 5Department of Life, Health and Environmental Sciences, University of L’Aquila, 67100 L’Aquila, Italy

**Keywords:** SARS-CoV-2, fragile workers, occupational health and safety, occupational physician

## Abstract

Background: The SARS-CoV-2 pandemic still represents a public health emergency that affects workplaces and forces employers to develop technical, organizational and procedural measures safeguarding workers’ health, particularly ‘fragile’ ones. This research aimed to assess employers’ adherence to the emergency measures planned by the Italian government to hinder COVID-19 during Autumn 2022. Methods: A cross-sectional study was conducted in Autumn 2022, with an 18-item questionnaire derived from the Italian State’s governmental indications, sent by email to 51 companies of Marsica and Peligna Valley, L’Aquila, Southern Italy. Results: A total of 20 recruited companies (65% were micro-enterprises, belonging to the food and financial sector) responded to the questionnaire within a mean time of 18 days (±11.64), which was lower for medium- and large-sized companies as well as for banking sector ones (*p* < 0.05). As regards intervention strategies, sanitization (92.7% of positive answers) and specific training (83.3%) showed almost full compliance in contrast to working organization (47.5%) and social distancing (61.7%). The companies that reported managing fragility (50%) belong almost exclusively to the banking sector, with predominantly office-based tasks. Conclusions: The study provided insight into critical issues relating to compliance with national legislative directives and the crucial role of occupational physicians as global advisors for all workplaces.

## 1. Introduction

The ongoing SARS-CoV-2 pandemic represents a public health emergency that has affected daily social life, including workplaces. Since the declaration of COVID-19 as a pandemic by the World Health Organization (WHO) from 11 March 2020 until 14 March 2022, more than 637 million confirmed COVID-19 cases and more than 6.6 million deaths have been reported [1]. Several viral variants influenced the SARS-CoV-2 circulation in non-simultaneous waves worldwide [2]. Legislators in various countries quickly produced many regulations to counter and contain the viral spread to help struggling occupations, involving restrictions and forced work closures. The first and invasive containment measure adopted by countries was undoubtedly the lockdown, the only measure that reduced COVID-19 mortality during the pre-vaccination era. Italy also adopted lockdown as the first containment measure, introducing less stringent measures to modulate the effects on occupational health, public health and the economy. Other containment measures are represented by contact tracing and molecular tests on nasopharyngeal swabs [3]. In Italy, from the start of the pandemic to 23 November 2022, 24,410,365 cases of COVID-19 were reported, of which 178,300 died. The vaccination campaign in Italy started on 27 December 2020; by the end of November 2022, high vaccination coverage levels were evident, with more than 142,000,000 administrated doses. Despite the vaccine coverage, from August 2021 to November 2022, re-infection cases amounting to 7.5% of the total number of known cases were reported. The latest rapid survey on the prevalence and distribution of SARS-CoV-2 variants in Italy showed that BA.5 remains predominant, with a national prevalence of 91.5% and regional/PPAA frequencies of over 81.8%. The highest incidence rate over the period analyzed was in the 50–59 age group (484 cases per 100,000), with a median age at diagnosis of 53 years [4]. The considerable difficulties in health management during the tumultuous evolution of the COVID-19 emergency led to pressing Italian legislation. As a consequence, a large number of decrees with increasing levels of restriction, explanatory notes, ordinances and circulars were issued by the government, the Ministry of Health, regions, and local health authorities, which have, however, suffered from a lack of homogeneity, only partly justified by territorial specificities. An important tool was undoubtedly the creation of a shared protocol among stakeholders aimed at regulating and unifying all the measures to combat and contain the spread of the COVID-19 virus in the workplace on the national territory, published for the first time in March 2020 and subsequently updated in April 2020, April 2021 and June 2022 [5]. This document was the source of inspiration for the present work. In addition, the Italian government immediately introduced protections for ‘fragile’ workers through decrees and ministerial operating circulars that provided updates and clarifications, which were gradually amended. Among the aspects that have been most taken into consideration are persons suffering from chronic diseases or with multimorbidity, or with states of congenital or acquired immunodepression, and public and private employees in possession of recognition of serious disability [6,7,8].

Occupational physicians (OPs) play a key role in workplace health management in Italy. In this pandemic, occupational physicians play a key role in monitoring workers’ health, acting to prevent the spread of pandemic outbreaks in the workplace and developing practical guidelines for returning to work. Specific attention by working organizations was paid to the controversial protection of ‘fragile workers’, characterized by confusing legislation, which pitted privacy against the protection of the weakest workers and mixed the competences of family doctors and forensic doctors, but often shifted the ultimate responsibility to OPs [5]. To date, considering the pandemic situation, the OPs had in fact gone beyond the boundaries drawn by current legislation, moving in emergency situations to respond to the new and pressing health and safety demands of the company system and the social community. The OPs’ experience during the pandemic was characterized by advice to key company figures or individual workers, a ‘point of reference’ fully involved in the operational management of the crisis. The evolution of the pandemic and consequently the continuous legislative changes did not in fact facilitate the work of the competent doctors, who needed to contextualize the plethora of regulations in each company according to specific needs while contributing firstly to the interruption of the chain of contagions and then to the safe return of workers to their workplaces [9,10].

This study took up the modalities of an earlier work by some authors [11] and was carried out in Abruzzo, a region in southern Italy with a population of 1.3 million and a population density of 121 inhabitants/km^2^. Through a survey conducted among a sample of companies in Abruzzo, Southern Italy, this research aimed to assess employers’ adherence to the emergency measures planned by the Italian government to hinder SARS-CoV-2 infection during the pandemic phase in autumn 2022, analyzing the balance between production needs and the regulatory requirements.

## 2. Methods

### 2.1. Study Design, Setting and Participants

A cross-sectional survey between November and December 2022 by the Department of Prevention of the Local Health Authority of L’Aquila, Italy, involving 51 local companies, was conducted. The enterprises were classified into micro (<10 employees), small (10–49), medium-sized (50–249) and large (>250) enterprises. As regards the working sector, all registered companies were classified according to the NACE Rev. 2 classification [12] and, for analysis purposes, also aggregated into four groups: the banking sector, food industry, manufacturing industry and ‘others’ group. The data were reported according to the instructions for Strengthening the Reporting of Observational Studies in Epidemiology [13].

### 2.2. Variables and Data Collection

The 18-item adopted questionnaire (Appendix A) was drafted on the basis of Regulatory COVID-19 Protocol developed by the Italian State’s government, considering the topics identified by the Italian Legislation and Ministerial Scientific-Technical Committee, aimed at the containment of viral spread [5]. The questionnaire was drawn up to investigate how companies complied with the law in responding to the pandemic emergency as a memorandum to stakeholders regarding regulatory dictates. The checklist items were grouped into four macro-areas, as shown in Figure 1.

Each item involves a closed question with three possible answers (‘YES’; ‘NO’; ‘Not applicable’) and a note field for details. The survey was conducted by e-mail, giving companies a seven-day deadline to respond.

### 2.3. Data Analysis

The responses were collected in Microsoft Excel spreadsheets (2016), and all variables were analysed using XLSTAT for Windows ver. 2022.1.2. Overall, the results were described in terms of mean (±SD) or median (25–75 IQR) frequencies and percentages for descriptive statistics and presented by tables and figures. An inferential statistical analysis was performed on some subgroups of data, subject to the assessment of the sample’s normality of the distributions by Shapiro–Wilk test (sample size < 30). In the case of normal distributions, parametric comparison tests (*t*-test or ANOVA) were used; for non-normal distributions, non-parametric tests (Mann–Whitney test for comparing two distributions, Kruskal–Wallis test for comparing more than two distributions) were used. A *p*-value of less than 0.05 was considered to be significant. Although highly relevant, the information from the notes could not be statistically analyzed because it is heterogeneous and merely descriptive.

### 2.4. Ethics

The research was conducted according to the ethical standards of the 1964 Declaration of Helsinki and subsequent amendments within the ordinary activities of the Department of Prevention of Local Authority Health in Abruzzo, Italy.

## 3. Results

Of the 51 companies recruited, 20 (39%), employing 2896 workers, fulfilled the questionnaire. All of the companies were in L’Aquila, Italy, particularly in the Peligna Valley and the Marsica area, and represented the main industrial production.

As regards company size, 65% of the companies responding to the questionnaire were micro-enterprises (mean workers, 7.20 ± 2.05), 40% were small-enterprises (27.13 ± 9.69), 25% were medium enterprises (156.60 ± 56.77) and only 10% were large enterprises (930 ± 579.83).

Table 1 details the sample, including working sectors and the size of the recruited companies.

### 3.1. Time for Response to the Survey

The mean and median times for answering the questionnaire were 18 days (±11.64) and 21 days (7–29.5, IQR), respectively. Only seven companies returned the questionnaire within the seven days’ deadline. As regards company size, the mean response time was 24.8 days (±9.34) for micro, 17.5 (±13.61) for small, 13.60 (±10.85) for medium and 14 (±9.90) for large companies. Medium- and large-sized companies answered more quickly on average than micro and small companies. The differences between the groups were statistically significant (*p* = 0.045).

For time response evaluation, we also considered working sector groups, as follows: banking sector (11, 12, 13, 15, 16), food industry (2, 3, 4, 10, 20), manufacturing industry (5, 7, 8, 9, 14, 17, 18, 19), and ‘others’ group (1, 6). The average response time of the banking sector (5.20 ± 3.42) was statistically lower than all groups (*p* = 0.019) (see Table 2). Companies belonging to the banking sector showed greater compliance in answering the questionnaire.

### 3.2. Companies’ Adherence to Anti-SARS-CoV-2 Prevention Measures

Working organization resulted in a macro-item with 51.7% of negative answers, the highest of the four intervention groups. For item #3 (has the non-essential business organization been reviewed regarding suspending/reducing specific production activities in this post-pandemic phase?), 75% of companies answered ‘NO’. Regarding the working sector, Banking companies showed a higher percentage of YES responses (63%), while NO responses were 60% for the food sector. Item #6 included a specific question on the management of fragile workers. Only half of the sample (10 companies) responded that they had to manage such workers; 35% of them used the open notes field to specify management. Regarding item #3, in the notes section, some companies commented that there was no need to remodel the company organization because of the low density of workers in large production areas.

Social distancing strategies were also downsized, with fewer restrictions imposed on moving within workplaces and meeting in common areas, as can be seen from the percentage of negative responses to items #9 and #10. The food and banking sectors reported 56% and 63% of YES responses, respectively. An exception was the possibility of conducting remote work meetings, which was still adopted by 80% of companies, and the use of face masks (100% positive responses).

Sanitization and specific training resulted in the approach with the highest number of positive responses, at 91.7% and 83.3%, respectively. All companies stated that they continue to provide ‘detergent and sanitizing gels’ (item #14), and almost all (19 out of 20) continued to provide ‘extraordinary sanitation/cleaning activities… due to COVID-19 risk’. Regarding specifying training, 95% of the companies maintained and updated information to workers on the risk from SARS-CoV-2 infection, as well as the on-site management of suspected cases (item #18).

Table 3 shows the mean distribution of the received responses according to each of the macro-items.

Furthermore, when analyzing the fulfilment of the ‘notes’ section, only 104 answers out of a total of 360 were provided. The most frequent comment concerned item #15, where 70% of the companies were mainly concerned about the products used for sanitization. The other comments were mainly found for work organization and social distancing, which were used to justify the high rate of negative answers.

The responses’ distribution in the study sample, and thus the adherence variation within the four SARS-CoV-2 preventive measures groups, is graphically represented in Figure 2.

## 4. Discussion

The SARS-CoV-2 pandemic has deregulated occupational activities worldwide, producing landmark changes in work habits [14]. The effective application of government policies to control pandemics requires public support. Han et al. showed that greater government reliance on COVID-19 control is associated with greater compliance with the recommended health behaviors and a slower decline over time. Potential factors modulating such adherence are better government organization in response to the pandemic, messages that are as consistent as possible, and greater perceived equity [15]. Among the various strategies promoted to limit the spread of COVID-19, many companies have been faced with new risk assessment models, often outside their own specific professional context, such as biological risk evaluation. In all these cases, the OPs’ clinical assessment for fitness for work requires close cooperation with employers, health and safety management, and human resources personnel.

This research work showed a reasonable overall adherence to the regulatory dictates by companies in the L’Aquila area, in Southern Italy, even in this last phase of COVID-19 spread, although with some blind spots in some preventive measures. After the first pandemic phase characterized by extreme containment measures (e.g., interruption of activity or removal of workers), the data presented so far show a reshaping of organizational models that takes into account the significant changes that this emergency has brought about. Several companies have activated an Internal Crisis Unit, as required by Italian law, in which all representatives of the company and trade union safety functions participate, and have set up a specific protocol for the management of any COVID-19-positive workers found in the workplace, in agreement with the local health authority.

Nonetheless, most of the enrolled companies complained about some obstacles to fully adhering to the ministerial recommendations due to the changed epidemiological conditions of the pandemic. Therefore, a scattering of responses was found in our sample, especially regarding work organization and social distancing, as represented in Figure 2. The large number of ‘NO’ results mainly concerned these two issues. One explanation could be that we are in a phase of economic recovery, whereby the reduction of the active workforce (through distanced working or leave/vacation, etc.), has been limited to extreme cases only (e.g., to protect fragile individuals with contraindications to vaccine prophylaxis). The reduction or partial modification of productive activity, which indirectly favored social withdrawal in the first pandemic phase, when effective means of individual protection (e.g., masks and vaccines) were unavailable, is no longer observable today. In our opinion, the ‘NO’ answers could also be justified mainly by the resumption of everyday habits, linked to the reduced COVID-19 risk perception due to SARS-CoV-2 virulence changes, as well as the almost complete vaccination coverage, which caused the issue to receive less attention even from trade union organizations.

Public knowledge and shared disclosure determine risk perception and public opinion on the disease [16]. Moreover, the reduction in the disease’s mortality rate has also played a critical role in changing the risk perception. Indeed, the Case Fatality Rate (CFR), i.e., the number of deaths in the population of diagnosed and notified cases, decreased from 19.6% at the start of the pandemic to 0.1% in September 2022. In one year (January 2021–January 2022), the CFR fell from 2.4% to 0.2%. The same downward trend was observed in the standardized CFR compared to the European and Italian population [4]. All these issues could represent the barriers to compliance in this cross-sectional experience. For the items of sanitization and specific training, a higher number of positive responses were recorded, indicating an almost total agreement with the instructions of the health authorities. In our opinion, the habit of being up-to-date about the pandemic, as well as easy access to masks and sanitizing devices with intensified cleaning phases on workplaces, are regularly retained as being beneficial. The masks’ use appears to be maintained by all working sectors, despite the epidemic downturn in the viral spread, in agreement with the significant improvement in the population’s choice and handling of face masks [17].

The authors recorded a long latency in responding to the questionnaire (median of 21 days), compared to previous experiences, where a median time to return by the deadline was reported [11]. This delay was observed in all sectors except banking. The long latency in receiving replies, which required several reminders, may be attributable to an increased intolerance of these issues from the employers, who are frequently subjected to supervisory activities by the local authorities responsible for monitoring adherence to the regulations. Overall, companies with a more significant number of workers showed relative ease in adopting flexible forms of work, which helped to adopt the regulatory protocol to contrast and contain the pandemic in the workplace; it is possible to infer from the data a sudden change and adaptability of their working models in the function of the epidemiological status of SARS-CoV-2 spread, despite the large number of workers employed. On the contrary, small- and medium-sized enterprises certainly experienced difficulties, mostly related to typical work activities, as for the food sector where, for example, smart working is inapplicable. In this case, the OP figure, as the leading medical advisor supporting companies, plays a key role in cases where the inapplicability of the operative proposals in the shared protocol were preponderant.

SARS-CoV-2 infected millions of workers, including vulnerable ones with pre-existing comorbidities (e.g., cardiac, neurological, neoplastic and metabolic diseases) or with ongoing therapies, such as immunosuppressive therapies [18]. In the wake of regulatory developments, the aspect of fragile workers was included in the survey and the critical issues inherent in the subject emerged. This is a surprising finding is the relationship of companies with fragile workers. In fact, the analysis of item #6 concerning the management of ‘fragile’ workers showed that 50% of the companies stated that they did not have any workers defined as fragile among their employees, and 90% were small enterprises. These companies mainly belong to the food sector, with no possibility of using flexible forms of work, due to the peculiarity of the work activity itself. Of the seven companies that used the notes field of the question, only three answered yes to specify the management of fragile workers. They explained that the OP had identified the workers according to the Ministry of Health Guidelines and that for some of them, forms of protected forced rest had been used (nine in one company, three in another), while for the others, the use of teleworking had been activated where possible. The companies that reported having had to manage individuals hypersusceptible to SARS-CoV-2 infection mostly belong to the banking sector.

Among the strengths of this study, the prospective generalizability of these data should not be underestimated. The data of this survey, coming from different work realities and highlighting any critical issues related to compliance with legislative directives, represented a fundamental employment observatory to support companies monitoring on-the-job applications of interventions aimed at contrasting the spread of SARS-CoV-2. The results of this research show good adherence to the regulatory dictates by the companies in the L’Aquila area right from the first phase of the spread of COVID-19. All the companies that in the first phase of the pandemic had preferred to resort to extreme containment measures (e.g., interruption of activity, national redundancy funds to support workers), had the time to safely resume suspended activities, plan a workplace remodeling and take into account the significant changes that this emergency entailed. The effectiveness of these preventive measures will be the subject of subsequent investigations with on-site evaluations of the companies examined.

Regardless, some limitations emerged from this study. First, the use of a questionnaire created by the authors, and inspired by the Italian legislation in consideration of the issues highlighted by the technical-scientific committee, has yet to be tested or validated. Therefore, this questionnaire is more like a checklist for an internal audit within companies to evaluate compliance to Italian regulatory requirements. Another issue is related to the companies’ enrollment, which was found to be lower than in the previous experiences of the same authors, even if the sample size of the involved workers in the analysis was comparable to previous work.

The experience gained in dealing with this emergency should consolidate the need to promote a comprehensive advisory figure for prevention aspects, acting in synergy with the territorial structures of public health, considering the case of this pandemic as the paradigm of the health problems osmosis with no boundaries between the work and the living environment. What certainly needs to be made clear is respect for the OPs’ competence area, which cannot be asked to encroach on tasks that belong to public authorities arbitrarily or to unwittingly extend health protocols to workers without scientific criteria or prior agreement [19,20].

This survey is to be considered as an update of a previous experience [11] with which it is interesting to compare how, with the same territory and work sectors concerned, despite the persistence of specific strongly recommended indications/suggestions and the persistence of a high viral prevalence, companies have relaxed their restrictive actions due to the changed characteristics of the virus and consequently its perception.

## 5. Conclusions

The data presented could represent a useful starting point to understand whether these measures can have possible social, economic and organizational impacts. The analysis of compliance and related obstacles faced by companies shows the resilience of companies that continue to modify their activities according to the pandemic’s epidemiological conditions. These experiences represent a wealth of corporate adaptability to preserve and maintain strategic know-how to protect the health and safety of workers during current and future pandemics.

## Figures and Tables

**Figure 1 ijerph-20-05105-f001:**
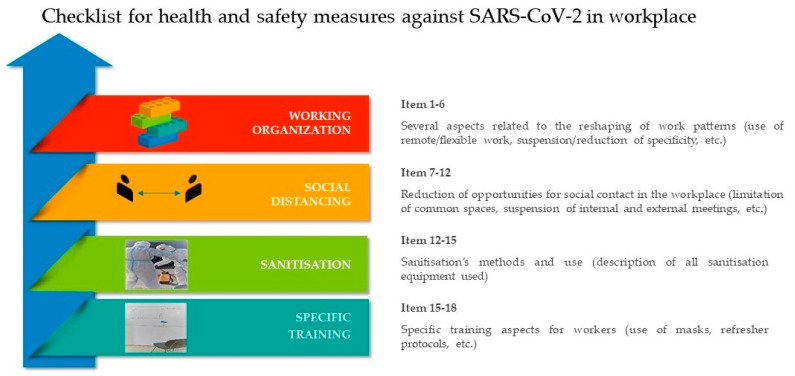
Representation of the four macro-items synthesis from the Italian Regulatory Shared Protocol.

**Figure 2 ijerph-20-05105-f002:**
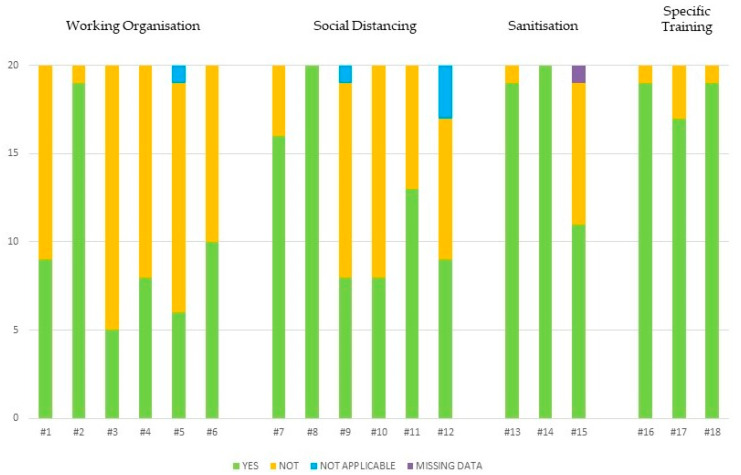
Graphical analysis of the responses’ distribution.

**Table 1 ijerph-20-05105-t001:** Sample size of the enrolled industries.

ID	Working Sector	No. Employer	Size Group
20	Food industry	4	micro
3	Food industry	7	
10	Food industry	7	
1	Waste collection, Treatment, Disposal	9	
2	Food industry	9	
9	Rubber and Plastics industry	12	small
6	Agricultural cultivation and Production of animal products	20	
13	Banking Sector	20	
8	Rubber and Plastics industry	28	
5	Rubber and Plastics industry	29	
15	Banking Sector	30	
14	Metal products manufacturing	35	
11	Banking Sector	43	
16	Banking Sector	90	medium
4	Food industry	120	
7	Textile industries	145	
12	Banking Sector	200	
19	Electronic Products industry	228	
17	Motor vehicles, Trailers and Semi-trailers manufacturing	520	large
18	Electronic Products industry	1340	

**Table 2 ijerph-20-05105-t002:** Framework of the survey’s return time.

	Survey’s Return Time (Days)	
Working Sector Group	m (±SD)	M (IQR)
Banking sector	5.20 (±3.42)	4 (4,7)
Food industry	22.80 (±8.73)	21 (21,31)
Manufacturing sector	19.75 (±11.5)	25 (7,29)
‘Others’	31 (±0)	31 (31,31)

m = mean; M = median; SD = standard deviation; IQR = interquartile range.

**Table 3 ijerph-20-05105-t003:** Percentages of the whole sample’s responses.

	Yes %	No %	N/A %	Missed %	*p*-Value
Working Organisation	47.50	51.67	0.83	0	<0.05
Social Distancing	61.67	35	3.33	0	
Sanitisation	91.70	7.47	0	0.83	
Specific Training	83.30	6.70	0	0	

## Data Availability

The data presented in this study are available upon request from the corresponding author.

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
