# Peer review of "SARS-CoV-2 Emergency Management in the ASL 1 Abruzzo Companies, Italy: An Autumn 2022 Cross-Sectional Investigation"

_ijerph, 2023, doi:10.3390/ijerph20065105_

Round 1

Reviewer 1 Report

Thank you very much for inviting me to review this interesting article. I wish to appreciate the research team for the present research. However, the article needs revisions to make it scientifically sound for the readers.

General comments:

The available evidence suggests that most of the countries controlled the COVID-19 situation, and companies are almost running at a pre-pandemic level. This study's results could help policymakers to make necessary emergency preparedness measures related to the occupation environment. The authors mentioned they followed the STROBE statement, and I suggest rechecking the STROBE statement and following the same throughout the article.

Title:

1.      Kindly capitalize "Sars -CoV-2" according to the international standard.

Introduction:

2.      Half the introduction states the global situation. However, I suggest giving more emphasis on the current situation in Italy.

Methodology:

3.      Even though the present study was submitted under the communication category, I suggest adding the psychometric properties of the ad-hoc questionnaire.

Discussion:

4.      Fairly well-written for communication. But the limitations of the present study are to be mentioned briefly.

Good luck!!

Author Response

please, see attached file

Reviewer 2 Report

- The abstract is almost uninformative. The Results are decriptive only

- The aim of the study must be clearly reported

- The authors stated they used an ad hoc questionnaire. However,  no psychometric issues are considered, especially in terms of reproducibility and internal consistency.

The methods are not adequate for a scientific paper, no inference is present according to the statistical analysis. As an example, data presented in Fifure 1 can be analysed using a hypotesis testing.

- The presentation of the Results is poor and somewhat confusing.

- The figures are not useful and can be easily replaced by a description in the text. Figure 1 is not easily readable and can be replaced by a table. In this case a chi-square test is applicable

Author Response

please, see attached file

Reviewer 3 Report

Thank you for allowing me to review this manuscript, compliance with regulations is an important topic. I would like to see more background on the regulations being verified. Not being from Italy I felt like a lot of context was missing that would help strengthen the paper. 

In the discussion, I would like to see more about why the "no" results were so high. The questionnaire had a notes section for employers to enter why they did not comply. This could be very rich information that would help employers and occupational health workers moving forward. Barriers to compliance would add significant impact to this manuscript. 

Author Response

Please, see attached file

Reviewer 4 Report

Line 172: A point (.) is needed after "emerged"

It is well thought-out

Author Response

Please, see attached file

Round 2

Reviewer 1 Report

Dear authors,

Thanks for making the necessary changes

With regards

Reviewer 2 Report

The manuscript has been much improved according to the reviewers' comments

Reviewer 3 Report

Thank you for allowing me to review this article again. You have addressed all of my concerns.